# Use of Drug Sensitisers to Improve Therapeutic Index in Cancer

**DOI:** 10.3390/pharmaceutics16070928

**Published:** 2024-07-11

**Authors:** Yu-Shan Chen, Enhui Jin, Philip J. Day

**Affiliations:** 1Division of Evolution, Infection and Genomics, Faculty of Biology, Medicine and Health, The University of Manchester, Manchester M13 9PL, UK; yu-shan.chen@postgrad.manchester.ac.uk (Y.-S.C.); enhui.jin@postgrad.manchester.ac.uk (E.J.); 2Department of Medicine, University of Cape Town, Cape Town 7925, South Africa

**Keywords:** therapeutic index, cancer, drug interaction, drug combination, polypharmacy, drug sensitisers, binary weapon, precision medicine

## Abstract

The clinical management of malignant tumours is challenging, often leading to severe adverse effects and death. Drug resistance (DR) antagonises the effectiveness of treatments, and increasing drug dosage can worsen the therapeutic index (TI). Current efforts to overcome DR predominantly involve the use of drug combinations, including applying multiple anti-cancerous drugs, employing drug sensitisers, which are chemical agents that enhance pharmacokinetics (PK), including the targeting of cellular pathways and regulating pertinent membrane transporters. While combining multiple compounds may lead to drug–drug interactions (DDI) or polypharmacy effect, the use of drug sensitisers permits rapid attainment of effective treatment dosages at the disease site to prevent early DR and minimise side effects and will reduce the chance of DDI as lower drug doses are required. This review highlights the essential use of TI in evaluating drug dosage for cancer treatment and discusses the lack of a unified standard for TI within the field. Commonly used benefit–risk assessment criteria are summarised, and the critical exploration of the current use of TI in the pharmaceutical industrial sector is included. Specifically, this review leads to the discussion of drug sensitisers to facilitate improved ratios of effective dose to toxic dose directly in humans. The combination of drug and sensitiser molecules might see additional benefits to rekindle those drugs that failed late-stage clinical trials by the removal of detrimental off-target activities through the use of lower drug doses. Drug combinations and employing drug sensitisers are potential means to combat DR. The evolution of drug combinations and polypharmacy on TI are reviewed. Notably, the novel binary weapon approach is introduced as a new opportunity to improve TI. This review emphasises the urgent need for a criterion to systematically evaluate drug safety and efficiency for practical implementation in the field.

## 1. Therapeutic Index and Its Use in the Pharmaceutical Sector

The pharmaceutical sector continues to face significant challenges in drug development. Approximately 90% of drug candidates do not achieve approval, especially failing during clinical trial stages III and IV, which is primarily attributed to insufficient clinical efficacy (40–50%), excessive toxicity (30%), unfavourable drug properties (10–15%), and commercial considerations (10%) [1,2,3,4]. Early identification of unsuitable drug candidates could mitigate costly late-stage clinical trial withdrawal, therefore, favourably shifting the balance towards drug safety and efficacy [5,6,7]. 

Drugs can have potential side effects [8], with drug dosage being positively linked to increased off-target (toxic) interactions. Despite this situation, drugs with benefits that outweigh the known risks, sometimes with caution, receive approval [9]. The Food and Drug Administration (FDA) has not prescribed any specific criteria for assessing drug efficacy and safety. Instead, the case-specific evaluation for each drug application via a benefit–risk framework (BRF) is usually conducted [10,11,12]. In addition, the FDA has recently been encouraging the quantitative benefit–risk assessment (qBRA). In this context, the multi-criteria decision analysis (MCDA) method, suggested by the European Medicine Agency (EMA), is used to support decision-making especially when traditional BRA is marginal [13]. While indexes represent straightforward approaches for quantifying both drug safety and efficacy, as shown in Table 1, the pharmaceutical industry still relies on these measures, with the therapeutic index (TI) being prominent.

The therapeutic index, also known as the therapeutic ratio, describes a range of drug doses that show higher efficacy of drug medication with no more than acceptable adverse effects [14]. With largely academic research-based origins, TI identifies lethal drug doses in animal models or toxicity in clinical studies as a therapeutic effect [15]. In general, TI is calculated by dividing TD_50_ (toxic dose) by ED_50_ (effective dose). TD_50_ is the dose of the target drug that will cause unacceptable effects in 50% of the population and replaced the ethically unacceptable lethal dose 50 (LD_50_) measurements in animal models. ED_50_ represents the dose of the target drug that produces a therapeutic effect in 50% of the population [16]. Establishing the likely TI could help reduce drug failure by eliminating unsuitable drug candidates at an earlier stage [15].

The FDA does not have criteria for TI; however, a high-value TI (above 10) is preferable for selecting universal drugs, and TI greater than 5 serves as a criterion for considering a drug candidate for further preclinical study. Nevertheless, a low TI (below 2), known as a narrow therapeutic index drug (NTI), is also available occasionally due to limited choices for some severe diseases [17,18], such as Warfarin. The therapeutic index quantifies a drug’s selectivity for its intended target and plays a vital role in determining the necessity for therapeutic drug monitoring (TDM) and dosage intervals, particularly in the case of NTI, like anti-cancer medications [19,20]. Antibiotics such as beta-lactams, or remifentanil (TI = 33,000), have a wide margin of safety and do not require precise drug monitoring [21,22]. Despite the potential extensive application of TI in assessing drug candidates and informing decision-making, the complexities of TI are often underestimated due to the varying interpretations of TI among different disciplines within drug development. For instance, TI calculations differ between preclinical animal and human tests, with TD_50_ replacing LD_50_ in clinical trials. Furthermore, pharmacological effects and toxicity are determined by the drug pharmacokinetics (PK), including absorption, distribution, metabolism, and excretion (ADME), and exposure of the specific tissue to the drug rather than the dosage; hence, plasma exposure is usually used as a surrogate for tissue exposure in calculating TI. Moreover, patients will usually be treated with multiple dosages; hence, TI sometimes is calculated using the drug exposure at a steady state rather than after the first dosage [23,24,25]. 

Several factors can affect the TI including the following: (1) Drug delivery methods and drug formulation. Both can significantly influence drug uptake, which in turn affects drug efficacy and toxicity. (2) Interindividual variability. This includes age, gender, genetics, weight, ethnicity, and disease history which affect drug metabolism and elimination. (3) Environmental factors including daily lifestyle, nutritional status, disease status, existence of co-morbidities, and diet can also impact drug metabolism and excretion, hence varying TI. (4) Drug interactions can lead to variation in PK and pharmacodynamics (PD), subsequently affecting the TI value [15,26,27]. 

However, in a practical clinical context, all these factors and intraindividual variation make comprehensive utility difficult to achieve in practice. Moreover, several limitations are associated with TI calculations. Firstly, measuring TD_50_ can be challenging (Figure 1A). Toxicity may be unpredictable, delayed, or unrelated to the dose, rendering comparisons between different drugs difficult due to differing endpoints. Secondly, assessing drug exposure in human tissues is typically challenging, ergo plasma exposure is often utilised as a surrogate for tissue exposure. However, this approach is limited to drugs with adequate membrane permeability. Nonetheless, it is not frequently employed as a guide for clinical therapy, primarily due to cost and resource constraints and because it may not consistently align with drug efficacy, particularly when considering interindividual variability. Mostly, drugs are assessed through direct and simple monitoring of the objective endpoints, subjective endpoints, and adverse reactions, rather than relying on TI measurements (Figure 1B). Thirdly, the applicability of TI is restricted by its inability to account for rare but severe toxicities that are detectable only in large populations and, therefore, only present in late-stage clinical trials, such as the withdrawn drug pergolide to treat Parkinson’s disease [28]. Additionally, if the quantitative efficacy curve of a drug does not align with its dosage-toxicity curve, the TI value may not fully reflect the drug’s safety (Figure 2). Lastly, TI does not consider drug interactions or synergistic effects, which may be relevant in real-life clinical scenarios.

A similar criterion, the therapeutic window (TW), is also widely used, which is affected by several factors: (1) the dosage range of drugs between a minimum dose of effective concentration and minimum toxic concentration, (2) the ability to distinguish the main targets and the non-targets, (3) the quality of the manufactured drug, (4) drug PK, and (5) individual cases. It can be applied in generic drugs, immunosuppressants, transplantation, cytotoxic drugs with antibody conjugates, and CRISPR/Cas9 technology [29,30]. 

Overall, the evaluation and application of TI in drug development are critical and complex. Achieving a balance between drug safety and efficacy requires an understanding of the limitations and complexities in TI calculations [5]. While in practice, the drug BRA often becomes a unique and case-specific undertaking, the specific clinical behaviour of drugs, which can also be a function of disease, has hampered the widespread adoption of TI. A new strategy of applying TI in practice is suggested to improve the specificity and efficacy of clinical medications. 

The primary medical models in use include evidence-based medicine, translational medicine, and precision medicine [31]. The implementation of precision medicine relies on the methods of evidence-based medicine, grounded in the principles of systems biology and multi-omics, with increasing input from engineering and modelling via synthetic biology [31,32,33,34]. Drug treatments are applied in a personalised manner, tailored to the individual’s genetic profile, living environment, and lifestyle. Precision medicine is currently focused on complex polygenic diseases such as cancer and diabetes, as well as rare diseases. It provides solutions to issues including uncharacterised targets and a lack of drug specificity in cancer immunotherapy [35,36,37,38,39]. Currently, several achievements have been made in implementing precision cancer medicine including the identification of new diagnostic biomarkers, integration of multi-omics, increased drug portfolio, long-term cost reduction of sequencing, data harmonization, and setup of prospective registry trials [40,41,42,43]. Given the complexity of the widespread utility of TI, its application in precision medicine is considered to be more accurate than using a universal standard for every patient, since the genetic profile and individual information including body weight, lifestyle, and disease history can then be applied to TI calculation, and the precise dosing regimen specified for the individual will lead to better clinical outcomes. However, challenges in precision medicine include difficulties in follow-up monitoring, the requirement of highly dynamic diagnostic technologies, increasing complexity, and analysis of real-world data proving too complex for detailed analyses [44].

## 2. Strategies to Improve Therapeutic Index in Cancer Treatment

Cancers affect around 50% of the UK population and are the focus of study for disease mechanisms in academia and efficacious product development within the pharmaceutical industry [45,46]. Both sectors converge on clinical trials and the identification of clinical staging that relates the presentation of the disease to risk and disease severity. However, drug resistance (DR) presents a significant challenge in cancer treatment and impairs clinical benefit [47]. The drug-resistant phenotype usually occurs in cancer cells due to (1) active expression of efflux transporters (drug efflux) [48], (2) low expression of influx transporters (drug influx) [49], (3) alteration of drug targets [50], (4) inhibition of related pathways [49], and (5) interference in the interaction between drugs and their targets [51]. Several strategies to overcome DR include refining drug delivery systems (DDS) [49,52], applying multiple anti-cancerous drugs (drug combination) [53,54], and employing drug sensitisers [55]. This section aims to review procedures developed to counteract mechanisms that limit drug uptake and action. 

### 2.1. Modification of Drug Delivery System

Drug delivery is a process of administrating pharmaceutical materials to reach designated therapeutic targets [56]. A drug delivery system (DDS) is a combination of physical/chemical/biological techniques to direct the controlled release and delivery of compounds, including antibodies, vaccines, drugs, peptides, proteins, and enzymes, aiming to minimise the off-target localisations and increase drug durability during transportation [57].

As shown in Table 2, several DDSs have been well established, with oral pills and injections being the most predominant methods of administration. Data from the US Food and Drug Administration (FDA) indicate annual sales of US dollar 2.45 billion for aspirin tablets and over 10 billion injections administered worldwide. Oral drugs are widely used due to their convenience and ease of use. To improve drug PK and therapeutic outcomes, various DDSs are applied to treat various diseases. Nevertheless, imprecise delivery that leads to side effects poses a significant challenge for clinical treatment. Therefore, effective increases in drug specificity via an appropriate delivery system are essential [58].

### 2.2. Administration of Multiple Anti-Cancerous Drugs

Several methods have been developed to combat cancer, including surgery, radiation therapy, cryotherapy, and the use of anti-cancer drugs. Among these, drug treatment emerges as the leading approach, classified into four main categories: (1) chemotherapy, (2) targeted therapy, (3) hormonal therapy, and (4) immunotherapy [80,81], as indicated in Table 3. 

From the various anti-cancer drug types summarised above, single-drug treatments that only target one pathway usually show limited drug efficiency and rapidly trigger DR [127,128]. These therapies are invariably insufficient for treating complex and multifactorial diseases such as malignant tumours, central nervous system (CNS) disorders, and immune disorders [129]. This contributes to high attrition rates and reduced cost recovery in drug discovery. Drug combination therapy was first established by Emil Frei et al. who conducted drug combination studies with anti-leukaemic agents in paediatric acute leukaemia patients [130,131]. Nowadays, the utilisation of multiple drugs is a common approach when addressing complex diseases or combating DR. This includes drug synergism when drugs that individually yield similar effects exhibit significantly enhanced outcomes [132]. 

There are several advantages of employing drug combinations, including maximising drug efficacy by targeting different mechanisms, reducing DR, and increasing the cost effectiveness of the drug development process, particularly when repurposing FDA-approved drugs for cancer treatment [133,134,135]. 

Several methods to conduct multiple drug combinations have been developed and applied to medical and surgical treatments: 

#### 2.2.1. Combinations of Drugs Targeting Different Pathways

To enhance drug efficiency, multiple drugs that target different disease-related proteins and pathways are applied to patients. Nevertheless, the treatment may also simultaneously lead to additional toxicity and adverse side effects. Various anti-cancer agents designed on natural compounds have demonstrated multiple functions across different pathways due to their inherent characteristics. For instance, (1) ruscogenin, an anti-inflammatory and anti-thrombotic steroid found in *Ruscus aculeatus* (a traditional medicine used as diuretics and treatment for urinary system disorders) [136,137], shows a decreased pattern in regulating MMP2, MMP9, VEGF, and HIF-1α that could suppress the metastasis of hepatocellular carcinoma [138], and (2) plumbagin, an extract from the medicinal plant *Plumbago zeylanica* (one of the oldest herbs in Ayurveda including functions such as anti-inflammatory and anti-tumour [139]) can down-regulate tumour cell growth by inhibiting the mTOR/PI3K/Akt pathway and EMT transition in prostate cancer [140], in addition, inducing ROS production for specific cell apoptosis [141,142]. Some advanced computational-based methods (such as chemogenomics, proteochemometric modelling, target fishing, and system pharmacology) are also applied for further investigation [143]. 

#### 2.2.2. Modification of Drug Delivery into a Multi-Agent Delivery System

DDS can be categorised according to different methods or materials. By increasing the specificity of the delivery system, drug efficiency is considered the main factor. Hence, most DDS are designed to be single-drug delivery, which is not fully relevant for combination drug delivery. The development of advanced equipment and compounds as a multi-drug delivery system (MDDS) compensates for the disadvantage of a single-drug delivery approach which often has an uncontrollable drug release and ergo overdose that could cause adverse effects. For instance, the titania nanotube (TNT) array is a prototype for self-automated electrochemical equipment suitable for implantable drug delivery with its sequential drug release [144]; in addition, biophysical materials such as polymer micelles, polymer-drug conjugates, polymeric nanoparticles, hydrogels, and liposomes are also designed for drug delivery for complicating diseases [145]. 

#### 2.2.3. Combination of Different Therapies 

Even if monotherapy is commonly used to treat numerous diseases, especially malignant cancers, the complexity of the human body can rarely be treated via a single therapy [146]. To overcome the challenges caused by drug resistance and low drug efficiency, therapies such as immunotherapy, radiotherapy, surgery, and photodynamic therapy are usually conducted together to treat one single disorder [147].

#### 2.2.4. Drug Interactions and Polypharmacy in Cancer

Despite the benefits of higher efficacy by combining drugs, drug interactions can arise when multiple drugs [148,149] are prescribed to a patient. The potential drug interactions can profoundly influence TI by altering both drug efficacy and toxicity, and can be classified into PK interactions, PD interactions, pharmaceutical interactions, and herbal interactions [150].

##### Pharmacokinetic Interactions

Cancer treatment, especially chemotherapy and radiotherapy, can significantly weaken the immune system, making patients more susceptible to fungal infections [151]. Hence, antifungal drugs such as ketoconazole may be used in co-treatment with the chemotherapy drug, cyclophosphamide. However, this potent inhibitor of the CYP3A4 enzyme suppresses the concentration of the cyclophosphamide metabolite by inhibiting both CYP2C9 and p-glycoprotein, leading to an elevated plasma drug level and subsequently higher toxicity [152]. This interaction is known as a PK interaction which is defined as the alteration of drug ADME, with the subsequent effect on drug plasma levels, drug efficacy, and toxicity profiles.

##### Pharmacodynamic Interactions

Drugs masking each other’s effects directly at the site of action or through common physiological pathways are termed PD interactions. For example, patients undergoing cancer treatment often suffer from body pain as a common side effect; hence, pain relief is commonly co-administrated with anti-cancerous drugs. The methotrexate used in leukaemia treatment can lead to high-risk interactions when combined with non-steroidal anti-inflammation drugs (NSAIDs). NSAIDs, often used as pain relief, compete with methotrexate for the same protein binding sites. The enhanced plasma level of active methotrexate often subsequently results in side effects including liver injury and severe bone marrow suppression [153]. 

##### Other Interactions

Except for PK and PD interactions, pharmaceutical interactions may occur when drugs influence each other’s physical or chemical properties at the level of drug formulation. For example, Erlotinib, a tyrosine kinase inhibitor used for treating non-small cell lung cancer, has its absorption affected by proton pump inhibitors (PPIs), through increased gastric pH and reduced drug solubility [154]. In addition, the rising use of natural supplements may heighten the significance of interactions between herbal medicines and drugs. For instance, St. John’s Wort (SJW), a herbal treatment for depression, can interfere with imatinib, a tyrosine kinase inhibitor used to manage chronic myeloid leukaemia. SJW enhances the activity of the enzyme CYP3A4, which metabolises imatinib, resulting in faster clearance of imatinib from the body [155]. 

However, polypharmacy can arise when multiple drugs, usually above five [148,149], are prescribed to a patient. This scenario often occurs in the elderly since they often suffer from age-related cancers, decreased renal and hepatic functions, and geriatric syndromes which increases the risk of multimorbidity and mortality [156]. According to a study in 2020, the prevalence of polypharmacy was 44% among 1,742,336 elderly adults tested [157]; therefore, improvements of TI in multiple drug treatments are essential to decrease the appearance of polypharmacy. 

Chemotherapeutic agents often require extra supportive components to manage the therapy process [158]. According to the reports from Globocan data in 2020, cancer has affected more than 60% of the elderly people in Europe [159]. Adverse events from chemotherapy agents such as nausea, vomiting, malnutrition, and myelosuppression would also increase the appearance of polypharmacy [160]. Polypharmacy is highly associated with chronic diseases such as hypertension (85%), diabetes (86.7%), and cardiovascular-related issues in elderly patients with cancer [161], largely due to their correlation with ageing. The primary concern with polypharmacy arises from inappropriate medication use, multiple prescriptions, low adherence, and underuse of drugs. Multiple drug interactions raise the risk of altering serum albumin levels, essential for drug delivery and efficacy in the circulatory blood system, eventually leading to cytotoxicity and new symptoms due to adverse events [162,163]. 

### 2.3. Drug Repurposing

Drug repurposing is a method that seeks new uses for existing drugs, which can be rapidly validated in clinical trials due to their well-studied functional activities and toxicity. Mostly, these drugs when used in isolation failed to reach satisfying clinical outcomes due to the lack of efficiency. Candidate drugs can re-enter clinical trials by repurposing the drugs to bind with more specific targets [164]. In the last decade, around one-third of the current drugs involved in drug repurposing generated 25% of the annual income for the pharmaceutical sector [165,166]. Tumours often involve multiple symptomatic phenotypes and pathogenetic mechanisms, making combined drug repurposing a promising approach that has been developed and applied in anti-cancer drug research.

Key examples of drug repurposing are as follows: (1) the use of mebendazole, an anti-hookworm medication, now recognised as a multi-targeted kinase inhibitor potentially more effective and with a preferred side effect profile for cancer treatment than existing kinase inhibitors; (2) sildenafil was repurposed into a pulmonary hypertension drug when it was originally designed for hypertension [167]; (3) tadalafil can inhibit myeloid-derived suppressor cells in patients when it used to be a PDR-5 inhibitor for erectile dysfunction; and (4) propranolol, inhibiting the metastasis and proliferation angiosarcoma cells, was a hypertension beta-blocker [168]. Three strategies for repurposing current drugs are shown in Table 4, including knowledge-based, signature-based, and phenotype-based repurposing [169].

### 2.4. Application of Drug Sensitising Agents 

Unlike the repurposing of drugs, a means to improve TI with a reduced drug dose and to simultaneously resurrect drugs that previously failed because of a too-high ED_50_ is through the application of drug synergy agents [177]. Drug synergy agents act to increase patient sensitivity to a drug and do not have direct biological activity to treat disease but can help enhance drug efficacy, reduce toxicity, and overcome drug resistance. This approach maximises therapeutic benefits while minimising adverse effects and primarily aims to increase the TI of drug treatments. Table 5 indicates increasing (⬆️) and decreasing (⬇️) impacts on TI. 

#### 2.4.1. Piperine

Piperine is an insoluble solid alkaloid produced by *Piper* species such as *Piper nigrum* L. (black pepper) and *Piper longum* L. (long pepper) [178] and has been demonstrated to enhance the bioavailability and pharmacokinetic properties when co-administrating with anti-tumour drugs. For instance, (1) a combination of rapamycin and piperine is proven to increase its bioavailability by 60% to cure human tuberculosis by enhancing autophagy activity. Via inhibition of Ser/Thr kinase mammalian target of rapamycin (mTOR) and maturation of the mycobacterial phagosome, the survival of *Mycobacterium tuberculosis* is inhibited [179,180]. (2) Nevirapine is a potent non-nucleoside inhibitor of HIV-1 reverse transcriptase mainly applied to HIV-1 infection. The co-administration of piperine and Nevirapine increases the bioavailability of Nevirapine with very few adverse effects [181]. (3) 5-FU, a common analogue of fluoropyrimidine chemotherapy drug inactivating thymidylate synthase (TS), leads to the inhibition of DNA replication and RNA synthesis in tumour cells [182]. Piperine increases the toxicity and bioavailability of 5-FU to suppress tumour growth by decreasing the half-life of drugs and expression of related efflux transporters (P-gp, MRP1, and BCRP) [183]. 

#### 2.4.2. Amifostine

Amifostine (WR-2721), a broad-spectrum cytoprotective agent, protects against mutagenic and carcinogenic damage by chemotherapy and radiation therapy in normal tissues without diminishing the anti-tumour activity [184]. Amifostine could also minimise the side effects by using higher antineoplastic doses of cisplatin chemotherapy in patients. The cytoprotecting mechanism of amifostine is by its active metabolite WR-1065, a thiol compound with scavenger activity against free radicals: (1)Radiation therapy: The oxidation tension and haemoglobin saturation caused by radiation therapy induce the oxidation of WR-1065 and HIF activation in normal tissues, activating the cytoprotection [185].(2)Chemotherapy: Several clinical studies suggest that amifostine does not interfere with the antineoplastic efficacy of different chemotherapeutic agents. The main function of amifostine is to protect the normal tissues under high doses of cisplatin in melanoma patients [185]. In addition, another study also points out that amifostine could further prolong the half-life of platinum-based chemotherapy which could increase the exposure of tumours to chemotherapeutic agents [186].

#### 2.4.3. Tariquidar

Tariquidar, a P-gp drug efflux pump inhibitor, works as an adjuvant against multidrug resistance in cancer undergoing clinical research [187]. With high potency and specificity to P-gp and low impact on common chemotherapeutic agents, tariquidar may result in accumulating intracellular concentrations of anti-cancer drugs [188].

Docetaxel/Paclitaxel: The combination of docetaxel/paclitaxel and tariquidar can increase the therapeutic efficacy of docetaxel/paclitaxel by increasing the specificity and the water solubility of various drugs and reducing drug resistance by efflux transporters in breast, non-small cell lung, ovarian, and prostate cancers [189,190].

#### 2.4.4. Binary Weapon

Numerous drugs have failed clinical trials due to their off-target effects and low drug efficiency; hence, there is a growing interest in optimising those validated drugs but with poor TIs. Here, a new approach termed binary weapons (BWs) is introduced which works similarly to drug sensitisers. Binary weapons lack toxicity yet show potential to enhance gemcitabine killing in pancreatic cancer cells via co-treatment [177]. In addition, the efflux transporter ABCC10 was found to be down-regulated with the addition of BW, highlighting the potential of BWs to inhibit efflux transporters [191]. Similar criteria for selecting BWs are established to develop new drugs for malaria [192]. According to the BW approach, the application of a small chemical fragment can potentially revive drugs that previously failed due to off-target effects and high toxicity. This strategy contributes to the development of more effective therapies, ultimately benefiting patients [191]. 

**Table 5 pharmaceutics-16-00928-t005:** The use of compound synergy to increase chemosensitivity in cancer therapy.

Drug Sensitiser	Co-Treated Drug	Description	Current Clinical Status	Functions	Effect on TI	Refs.
Piperine	(1) Rapamycin	(1) Activate the autophagy pathway.	(1) FDA-approved	Bioavailability enhancers	ED_50_ ⬇️	[180,193]
	(2) Nevirapine	(2) Inhibit CYP450 and UDP glucuronyl transferase.	(2) Not approved due to limited clinical trials; few ongoing clinical trials			[194]
	(3) 5-FU	(3) Shorten the half-life of the drug.	(3) Under clinical investigation			[195]
Amifostine	Cisplatin	Cytoprotective agent used in chemotherapy.	FDA-approved, and it is used in clinic with cisplatin	Toxicity reducers	TD_50_ ⬆️	[196,197]
Tariquidar	Docetaxel, Paclitaxel	Inhibit P-gp efflux transporter to enhance drug efficacy.	Under clinical investigation	Efflux inhibitors	ED_50_ ⬇️	[187]
Binary weapon	Gemcitabine	Enhances the gemcitabine toxicity to pancreatic cancer cells by co-treatment but non-toxic by sole treatment.	Laboratory evidence	Enhance anti-cancer drug efficacy and specific to cancer cells	ED_50_ ⬇️	[191]
Bovine lactoferrin	Cisplatin	Sensitise Cis-anti-neoplastic potency.	No clinical data	Enhance drug efficacy, immunomodulators	ED_50_ ⬇️	[198]
Fedratinib	Vincristine	Fedratinib is a JAK2 inhibitor that sensitises P-gp-overexpressing drug-resistant cancer cells.	FDA-approved	Inhibits P-gp activity, inducing cytotoxicity and apoptosis in drug-resistant cancer cells	ED_50_ ⬇️	[199]
MG132	Idarubicin	Inhibit NF-κB regulations.	Under clinical investigation	Inhibition of NF-κB induces apoptosis of leukaemic stem cells and leaves normal cells viable	ED_50_ ⬇️	[200,201]
Urolithin A (UroA)/UAS03 (UroA analogue)	5-FU	Inhibit cancer cell viability, proliferation, and invasion in colon cancer cells.	Laboratory evidence	Enhance drug efficiency and inhibit the expression of related efflux transporters	ED_50_ ⬇️	[202]

### 2.5. Educating Patients

Patient behaviour has a significant impact on the TI, particularly for drugs with NTIs. Adherence, which refers to the extent to which a patient’s actions follow prescriber instructions, is essential for achieving optimal clinical outcomes [203]. It is reported that patients with good adherence to treatment have a lower rate of mortality and fewer hospitalisations compared to those who do not [204]. Moreover, it is found that adherence is significantly correlated with positive treatment outcomes, which makes the outcome predictable [205]. Non-adherence can manifest as frequent dosing, leading to toxic drug levels and adverse effects or missing doses, resulting in reduced drug efficacy and disease progression, and hence, lowering TI. Therefore, the development of pharmaceuticals within a DDS [206] offers a promising solution to many common impediments to adherence. An example is transdermal patches, which provide continuous release of medication over an extended period, eliminating the need for daily manual drug administration. This approach is widely used for hormone replacement therapy and the treatment in psychiatric illness [207]. Ensuring that patients understand their medications, maintaining open communication between patients and prescribers, and educating patients about their treatment are crucial for optimizing therapeutic outcomes and improving TI. Simplifying drug regimens and involving regular monitoring and follow-up can enhance the quality of care for patients receiving complex medications.

## 3. Conclusions and Future Perspectives

The therapeutic index has recently attracted much attention as it aims to evaluate drug efficiency and safety more accurately. Nevertheless, its implementation in cancer treatments can be challenging due to the difficulties in measuring TD_50_ and tissue exposure, detection of severe and rare toxicity, and unknown drug-drug interactions. To improve the efficiency of most failed drugs (improve TI), (1) modification of drug delivery systems, (2) multiple drug combinations, (3) drug repurposing, and (4) educating patients are continuously applied during drug developments. Moreover, the field is moving towards using simulation to predict drug behaviour. Machine learning algorithms are being reported to have the potential to predict drug efficacy and toxicity by building a predictive model, allowing for better personalised treatment [208].

From our previous studies [191], BWs are new materials with no known pharmacological activities and only show the advantage of increasing drug efficiency and specificity and lowering the drug dose during co-administrations. Given the reduced off-target potential with associated increased drug specificity, BWs could be the newest strategy to improve TI value for drug development and lessen drug-drug interactions through reduced drug dosage. In addition, reducing the number of drugs consumed and the use of polypills could also minimise the risks of toxicity and unknown DDI in polypharmacy and drug combination therapies. 

A long-term goal for drug development is to build precision medicine portfolios for different diseases, and the improvement of TI by BWs has increased the opportunities for more customised medication to help cancer patients receive more targeted clinical benefits in the future (Figure 3). 

## Figures and Tables

**Figure 1 pharmaceutics-16-00928-f001:**
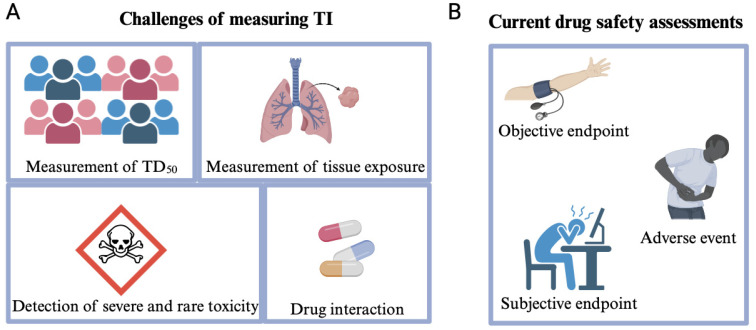
Challenges of measuring TI and current drug safety assessment methods. To provide a suitable drug administration to patients, physicians make several safety assessments. (**A**) However, there are some challenges to measuring TI due to unpredicted TD_50_, tissue exposure, limitations in detecting severe and rare toxicity, and unknown drug interactions. (**B**) Instead of using a specific criterion, drug safety is assessed by using an objective endpoint, subjective endpoint, and adverse event.

**Figure 2 pharmaceutics-16-00928-f002:**
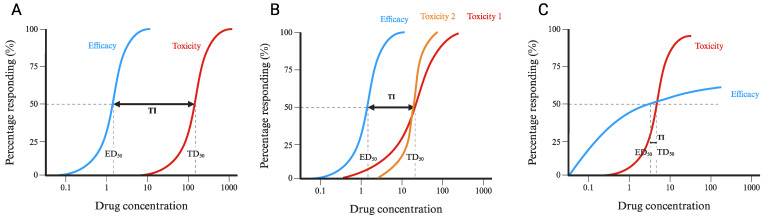
Dose-response curves used for TI calculations. The blue curve represents therapeutic efficacy, while the red and orange curves represent toxic effects. (**A**) represents the ideal dose-response curve. Drug concentration between the two curves, labelled as TI, are appropriate dosages that make the drug efficient but without severe side effects. The therapeutic index is then calculated using ED_50_ and TD_50_. In (**B**), toxicity curve 1 kills approximately 10% of the population at ED_50_, while toxicity curve 2 does not cause any deaths at ED_50_ but then rapidly becomes lethal after TD_50_. (**C**) shows when the efficacy curve does not align with the toxicity curve, and TI is not reliable.

**Figure 3 pharmaceutics-16-00928-f003:**
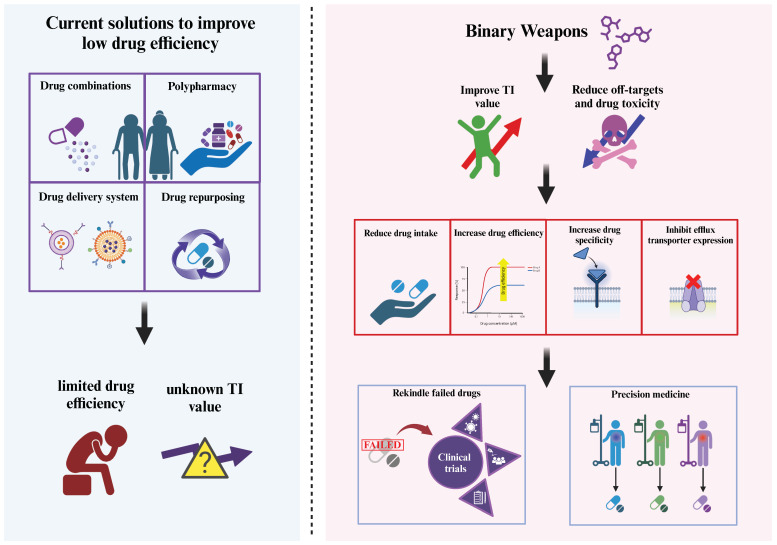
The advantages of applying binary weapons in drug administrations. The diagram describes the current solutions to improve low drug efficiency and the future perspectives of co-administrating BWs in clinical treatments. BWs could improve TI by increasing drug efficiency and reducing toxicity, leading to reduced drug intake, increased drug specificity, and overcoming drug resistance by inhibiting efflux transporters. The application of BWs provides opportunities for drugs that previously failed to re-enter clinical trials. Furthermore, using a reduced drug dosage while maintaining drug efficacy can pave the way for precision medicine, offering highly targeted treatment for a broad range of disease risk groups.

**Table 1 pharmaceutics-16-00928-t001:** Criteria used to assess drug safety and efficacy.

Criteria	Equation	Pros and Cons
Half-Maximal Effective Dose (ED_50_)	ED_50_ = Dose at which the drug produces a half-maximal effect	Reflects the relationship between therapeutic effect and acute toxicity but not chronic toxicity and allergenicity.
Lethal Dose (LD_50_)	LD_50_ = Dose that is lethal to 50% of the test population	Can evaluate the acute toxicity but does not reflect the chronic toxicity and carcinogenicity. LD_50_ may vary due to varying testing conditions.
Therapeutic Index (TI)	TI = LD_50_/ED_50_ = TD_50_/ED_50_	Widely used safety index but is not applicable to very rare or idiosyncratic adverse drug reactions.
Maximum Tolerated Dose (MTD)	The highest dose of a drug that can be administered without unacceptable toxicity	Usually used to assess chemotherapy drugs.
Therapeutic Window (TW)	TW = minimum toxic concentration (MTC)/minimum effective concentration (MEC)	Related to TI and is more flexible but lacks formal definition.
Margin of Safety (MoS)/Certain safety factor (CSF)	MoS/CSF = TD_1_/ED_99_	Patients will not be exposed to high risks. Can be seldom achieved in clinic.

**Table 2 pharmaceutics-16-00928-t002:** Comparison of different drug delivery systems ranked by clinical application prevalence.

Routes of Drug Administration	Working Model	Advantages	Disadvantages	Current Drugs	Refs.
Oral drug delivery	1. Small molecule delivery 2. Patient self-administration	1. Sustained and easy administration 2. Major method to establish patient compliance 3. Large surface area for mucosal layer attachment	1. Need to pass through GI system (multiple barriers) 2. Slow absorption 3. Degradation problems	1. Venlafaxine hydrochloride 2. Diltiazem 3. Indomethacin 4. Heparin	[59,60,61]
Injectable drug delivery	1. Protein and peptide delivery 2. Intravenous (IV), intramuscular (IM), intranasal (IN) 3. Induce immune response mechanisms	1. The highest bioavailability and the fastest effect 2. Acute and emergency responses	1. Needle-related pain, wounds, phobia	1. Glucagon-like peptide-1 2. Insulin 3. Superoxide dismutase 4. Hydrocodone (Vicodin)	[62,63,64,65]
Transdermal patch drug delivery	1. From skin layers to the blood circulatory system	1. Direct treatment away from GI system 2. Can maintain sustained drug level 3. Readily administered	1. Lower drug absorption level	1. Nitroglycerin 2. Nicotine 3. Scopolamine 4. Clonidine 5. Fentanyl 6. Testosterone	[66,67,68]
Ocular drug delivery	1. Deliver drugs to eyes against disorders related to vision 2. Formation: eye drop, eye implant	1. Easy administration and preparation 2. High patient convenience and compliance	1. Poor bioavailability 2. Low retention time 3. Side effect caused by high-frequency administration 4. Instability for dissolved drugs	1. Ocusert^®^ Pilo-20 2. Pilo-40 Ocular system	[69,70]
Pulmonary drug delivery	1. Inhalation of drugs via nebulisers or inhalers 2. Drugs that target the lungs	1. Rapid effects 2. High therapeutics due to large surface of lungs	1. Drug irritation to the lung 2. Limited drug dissolution 3. High drug clearance	1. Nebulisers 2. Pressurised metered dose inhalers (pMDIs) 3. Soft-mist inhalers 4. Dry powder inhalers (DPIs)	[71,72]
Implantable drug delivery	1. A reservoir surrounded by polymers or drug–polymer mixture 2. Passive delivery: use diffusion, osmosis, or gradients to control drug release 3. Active delivery: activate a pump to release drugs	1. Reduce dosing frequency 2. Increase patient compliance	1. Lack of systemic treatment 2. Undegradable, need a removal process	1. Vitrasert 2. Norplant^®^	[73,74,75]
Antibody drug conjugate delivery	1. Combination of clonal antibodies and drugs 2. Conjugation between target sites and antibodies	1. Highly toxic drug delivery (high specificity, low off-targets)	1. Poor tumour penetration (ex: hypoxic area) 2. Side effects in the non-target sites 3. Undesirable immune response (ex: Fc interaction)	1. Brentuximab vedotin 2. Trastuzumab emtansine (Kadcyla)	[76,77,78]
Polysaccharide-based hydrogel drug delivery	1. Use natural polymers to build beads 2. Suitable for peptides, proteins, DNA, and RNA	1. High biocompatibility and biodegradability 2. Cost efficiency 3. Ease of surface modification 4. Low toxicity 5. Rapid drug release	1. The quantity and homogeneity of drugs are limited (hydrophobic drugs) 2. Unable to bind with hard tissues (bone) 3. Difficult to sterilise	1. Poly (ethylene glycol)-diacrylate (PEGDA)	[56,79]

**Table 3 pharmaceutics-16-00928-t003:** Traditional and new classified anti-cancer drugs with clinical examples.

Classification of Anti-Cancer Drugs	Principle	Subtypes	Examples	Ref.
Chemotherapy	Interfere with tumour cell cycle, cell proliferation, and replication	(1) Alkylating agents	(1) Cyclophosphamide, chlormethine	[82,83]
		(2) Anti-metabolites	(2) 5-FU, 6-mercaptopurine, gemcitabine	[84]
		(3) Anti-tumour antibodies	(3) Atezolizumab, trastuzumab-deruxtecan	[85,86]
		(4) Topoisomerase inhibitors	(4) TOPI: camptothecin TOPII: doxorubicin	[87]
		(5) Tubulin-binding drugs	(5) Microtubule-stabilising: taxanes Microtubule-destabilising: vincristine	[88]
		(6) Antibiotics	(6) Bleomycin, daunorubicin, doxorubicin	[89,90]
		(7) Mitosis inhibitors	(7) Alisertib, ispinesib, GSK461364	[91]
Targeted therapy	Target specific proteins or genes related to cancer growth	(1) Receptor tyrosine kinase inhibitors	(1) Erlotinib, gefitnib, lapatnib, afatinib	[92,93]
		(2) Intracellular tyrosine inhibitors	(2) Imatnib, nilotnib, everlimus	[92,94]
		(3) DNA/RNA synthesis inhibitors	(3) Capecitabine, oxaliplatin	[95]
		(4) Topoisomerase I inhibitors	(4) Irinotecan, belotecan, topotecan	[96,97]
		(5) Proteasome inhibitors	(5) Bortzomib, ixazomib, carfilzomib	[98]
Hormonal therapy	Inhibit tumour growth dependent on hormones	(1) Steroids	(1) Dexamethasone, methylprednisolone	[99,100]
		(2) Anti-estrogens	(2) Tamoxifen, raloxifene, toremifene	[101,102,103]
		(3) Anti-androgens	(3) Bicalutamide, enzalutamide	[104]
		(4) LHRH conjugated drugs	(4) LHRH-paclitaxel, LHRH-prodigiosin	[105]
		(5) Anti-aromatase agents	(5) Exemestane, anastrozole,	[106,107]
Immunotherapy	Induce anti-tumour responses from the immune system	(1) Interferon	(1) IFNα-1a, IFNα-1b	[108]
		(2) Interleukin 2	(2) Aldesleukin	[109]
		(3) Vaccines	(3) Sipuleucel-T	[110]
		(4) Oncolytic virus therapy	(4) T-VEC	[111]
Others	(1) Disrupt energy production, essential cellular processes in mitochondria (2) Induce apoptosis pathways	Mitochondria-targeted anti-cancer drugs (mitocans)		
		(1) Hexokinase inhibitors	(1) 2-deoxyglucose, 3-bromopyruvate	[112,113]
		(2) Bcl-2/Bcl-xL mimetics	(2) Antimycin A, Gossypol, ABT-263	[114]
		(3) Thiol redox inhibitors	(3) Dichloroacetate, isothiocyanates	[115]
		(4) VDAC/ANT targeting drugs	(4) CD437, lonidamine	[116,117]
		(5) Electron transport chain targeting drugs	(5) Tamoxifen, MitoVES	[118,119]
		(6) Lipophilic cations targeting inner membrane	(6) MKT-077, Rhodamine-123	[120,121]
		(7) Drug targeting TCA cycle	(7) DCA, 3-bromopyruvate	[122]
		(8) Drug targeting mtDNA	(8) Vitamin K3, Mito VES	[123]
	Induce cells to produce specific proteins	mRNA drugs		
		(1) Vaccines	(1) mRNA-4157, pembrolizumab	[124]
		(2) Antibodies	(2) Anti-HER2	[125]
		(3) Antigen receptors	(3) Chimeric antigen receptor T cell therapy	[126]

**Table 4 pharmaceutics-16-00928-t004:** Three strategies for repurposing drugs.

Strategy	Concept	Pros and Cons	Examples	Refs.
Knowledge-based repurposing	Use the properties of drugs to predict the possibility of treating diseases: (1) Target-based repurposing (2) Pathway-based repurposing (3) Target-based repurposing	Pros: 1. Large scale prediction 2. Precise prediction 3. Time-efficient 4. Cost-effective Cons: 1. Only positive data can be found in knowledge database	1. CAS biomedical knowledge graph for COVID-19 2. L-type calcium channel blockers for cryptococcosis	[170,171,172,173]
Signature-based repurposing	A method to discover new off-targets or pathways. Genetic and molecular mechanisms are highly involved in the analysis.	Pros: 1. Identify new mechanisms of drugs Cons: 1. Only provide data for preliminary analysis	1. New candidates for atypical meningioma 2. New candidates for inflammatory bowel disease	[174,175]
Phenotype-based repurposing	A method for systemic approaches to detect human diseases, multiple independent screens for similar compounds and their potential for repurposing.	Pros: 1. Can predict extra adverse events Cons: 1. Might lead to compounds with poor pharmacokinetics	1. Electronic health records (EHRs) define the use of metformin in cancer treatment 2. Killing ability towards Toxoplasma by Pimozide and tamoxifen	[167,176]

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
