# Peer review of "Use of Drug Sensitisers to Improve Therapeutic Index in Cancer"

_pharmaceutics, 2024, doi:10.3390/pharmaceutics16070928_

Round 1

Reviewer 1 Report

Comments and Suggestions for Authors

The complexity and challenges associated with managing malignant tumors, particularly in the context of drug resistance and the need to enhance treatment effectiveness while minimizing adverse effects. The use of drug combinations and drug sensitizers to overcome DR and improve therapeutic outcomes in cancer treatment is a promising approach that warrants further exploration and development. The following details require attention:

1.      Improving the Therapeutic Index (TI) in the context of multiple drug treatments is crucial for managing polypharmacy and optimizing outcomes. Among, educating patients about their medications, including proper dosing, potential side effects, and the importance of adherence, is essential for promoting medication safety and effectiveness. Engaging patients in shared decision-making, encouraging open communication about their treatment experiences, and addressing any concerns or misconceptions can support medication compliance and optimize therapeutic outcomes.

2.      It is recommended that the authors add a description of the multiple drug combinations, modification of drug delivery systems, and drug repurposing to highlight their benefits.

3.      How to enhance the quality of care for patients with complex medication regimens?

Comments on the Quality of English Language

Minor editing of English language required

Author Response

Comment 1: Improving the Therapeutic Index (TI) in the context of multiple drug treatments is crucial for managing polypharmacy and optimizing outcomes. Among, educating patients about their medications, including proper dosing, potential side effects, and the importance of adherence, is essential for promoting medication safety and effectiveness. Engaging patients in shared decision-making, encouraging open communication about their treatment experiences, and addressing any concerns or misconceptions can support medication compliance and optimize therapeutic outcomes.

We thank reviewer 1 for their insightful comments which we have addressed below.

Reply 1: We added a new section related to educating patients from line 398 to 416.  

Comment 2: It is recommended that the authors add a description of the multiple drug combinations, modification of drug delivery systems, and drug repurposing to highlight their benefits.

Reply 2: We would like to clarify that chapter 2 (lines 164 to 416) relates to the points raised by comment 2, where the drug delivery systems are shown in table 2 (line 193) and the drug repurposing is mentioned in table 4 (line 320).  

Comment 3: How to enhance the quality of care for patients with complex medication regimens?

Reply 3: Lines 414 to 416 directly address comment 3. The purpose of this investigation is to demonstrate that the addition of binary weapon retains drug pharmacological effects at a lower drug concentration, and therefore, there is reduced opportunity for drug-drug interactions. In the abstract,  the statement has been mentioned from line 18 to 21, and a new comment has been added from line 431 to 434.    

Reviewer 2 Report

Comments and Suggestions for Authors

The therapeutic index (IT) is receiving considerable attention as it can provide a more accurate evaluation of drug effectiveness and safety. However, implementing it in cancer treatments can be challenging due to limitations in measuring TD50 or unpredictable drug toxicity from interactions. To improve the low efficiency of many failed drugs, drug development efforts focus on modifying drug delivery systems, using combination therapies, and repurposing existing drugs. Despite these efforts, a more accurate therapeutic index has yet to be established.

Manuscript strength:

It gives an overview of a relatively new topic, very well done Figures in terms of scientific content and quality of workmanship.

Weakness of the manuscript:

The authors could have written something about whether there is a mathematical prediction of the IT of the drug itself (based on the structural characteristics of the molecule-drug) or a binary mixture of two drugs or a drug and a modifier.

Suggested corrections:

The authors could have a chapter on how to calculate IT with some specific applications and whether there are predictive mathematical methods and if so what is their robustness and predictive power for IT before the drug enters clinical trials.

Author Response

Weakness of the manuscript:

The authors could have written something about whether there is a mathematical prediction of the IT of the drug itself (based on the structural characteristics of the molecule-drug) or a binary mixture of two drugs or a drug and a modifier.

Suggested corrections:

The authors could have a chapter on how to calculate IT with some specific applications and whether there are predictive mathematical methods and if so what is their robustness and predictive power for IT before the drug enters clinical trials.

We thank reviewer 2 for the useful comments, and for identifying the strengths and value of our novel studies. The weakness and suggested corrections identify by the reviewer both relate to the inclusion of mathematical tool to predict TI. We have made comments on lines 425 to 428 that machine learning algorithms will in the future be helpful in this regard.  

Reviewer 3 Report

Comments and Suggestions for Authors

The manuscript authored by Yu-Shan Chen et al.: ”Use of Drug Sensitizers to Improve Therapeutic Index in Cancer”, presents the importance of therapeutic index in evaluating drug dosage for cancer treatment and discusses the meaning of a unified standard for this in the field of cancer. Besides, the manuscript describes several factors that might influence the therapeutic index, but just few of them are further properly presented in text. The environmental factors such as diet and life style which can impact the drug metabolism are not discussed later in text.

The manuscript has in general well written sections which explain the importance of therapeutic index and presents several strategies to improve therapeutic index in cancer treatment.

Although the article aims to discuss the use of drug sensitizers for improved therapeutic index in cancer, as its title announces, only one short section is dedicated to this topic. In table 5, few examples of drug sensitizers are presented, but no further description of their mechanism of action can be find in text. This section should be extended and the home take idea of this manuscript should be clearer presented in conclusion and future perspective section. Also, few paragraphs about the environmental factors with impact on drug metabolism should be added.

Also, please revise the entire manuscript to check the typo (there are many different types of fonts and font sizes in text)

 Line 330: reference is missing

Line 338 typos

Author Response

The manuscript authored by Yu-Shan Chen et al.: ”Use of Drug Sensitizers to Improve Therapeutic Index in Cancer”, presents the importance of therapeutic index in evaluating drug dosage for cancer treatment and discusses the meaning of a unified standard for this in the field of cancer. Besides, the manuscript describes several factors that might influence the therapeutic index, but just few of them are further properly presented in text. The environmental factors such as diet and life style which can impact the drug metabolism are not discussed later in text.

Reply: Whilst environmental factors are briefly discussed (line 90 to 92), the focus of this article is the discussion of drug sensitizers to facilitate improved effective dose to toxic dose ratios directly in humans as indicated in the abstract lines 24 and 26. In addition, this review focuses on the strategies for improving TI through the use of drug sensitizers and not other factors such as diet and metabolism which we were not discussed in high details.  

The manuscript has in general well written sections which explain the importance of therapeutic index and presents several strategies to improve therapeutic index in cancer treatment.

Reply: We thank the reviewer for their kind comments. 

Although the article aims to discuss the use of drug sensitizers for improved therapeutic index in cancer, as its title announces, only one short section is dedicated to this topic. In table 5, few examples of drug sensitizers are presented, but no further description of their mechanism of action can be find in text. This section should be extended and the home take idea of this manuscript should be clearer presented in conclusion and future perspective section. Also, few paragraphs about the environmental factors with impact on drug metabolism should be added.

Reply: We thank the reviewer for these important suggestions which we have incorporated to improve our review, and have extended the content of table 5 and added 2.4.1 to 2.4.3 (line 341 to 383). 

Also, please revise the entire manuscript to check the typo (there are many different types of fonts and font sizes in text)

Reply: The script has been proof read.   

 Line 330: reference is missing

Reply: We added the reference which is on line 335. 

Line 338 typos

Reply: We corrected the typo on line 338 from "weapons" to  "weapons" which is now on line 389. 

Round 2

Reviewer 3 Report

Comments and Suggestions for Authors

Thank you to authors for responded to my comments and revised the manuscript. From my point of view,  this revised form of the manuscript could be published.